# Near-optimal sample compression
# for nearest neighbors

**Lee-Ad Gottlieb**
Department of Computer Science and Mathematics, Ariel University
Ariel, Israel. `leead@ariel.ac.il`

**Aryeh Kontorovich**
Computer Science Department, Ben Gurion University
Beer Sheva, Israel. `karyeh@cs.bgu.ac.il`

**Pinhas Nisnevitch**
Department of Computer Science and Mathematics, Ariel University
Ariel, Israel. `pinhasn@gmail.com`

## Abstract

We present the first sample compression algorithm for nearest neighbors with non-trivial performance guarantees. We complement these guarantees by demonstrating almost matching hardness lower bounds, which show that our bound is nearly optimal. Our result yields new insight into margin-based nearest neighbor classification in metric spaces and allows us to significantly sharpen and simplify existing bounds. Some encouraging empirical results are also presented.

## 1 Introduction

The nearest neighbor classifier for non-parametric classification is perhaps the most intuitive learning algorithm. It is apparently the earliest, having been introduced by Fix and Hodges in 1951 (technical report reprinted in [1]). In this model, the learner observes a sample $S$ of labeled points $(X, Y) = (X_i, Y_i)_{i \in [n]}$, where $X_i$ is a point in some metric space $\mathcal{X}$ and $Y_i \in \{1, -1\}$ is its label. Being a metric space, $\mathcal{X}$ is equipped with a distance function $d : \mathcal{X} \times \mathcal{X} \to \mathbb{R}$. Given a new unlabeled point $x \in \mathcal{X}$ to be classified, $x$ is assigned the same label as its nearest neighbor in $S$, which is $\mathrm{argmin}_{Y_i \in Y} d(x, X_i)$. Under mild regularity assumptions, the nearest neighbor classifier's expected error is asymptotically bounded by twice the Bayesian error, when the sample size tends to infinity [2].[1] These results have inspired a vast body of research on proximity-based classification (see [4, 5] for extensive background and [6] for a recent refinement of classic results). More recently, strong margin-dependent generalization bounds were obtained in [7], where the margin is the minimum distance between opposite labeled points in $S$.

In addition to provable generalization bounds, nearest neighbor (NN) classification enjoys several other advantages. These include simple evaluation on new data, immediate extension to multiclass labels, and minimal structural assumptions — it does not assume a Hilbertian or even a Banach space. However, the naive NN approach also has disadvantages. In particular, it requires storing the entire sample, which may be memory-intensive. Further, information-theoretic considerations show that exact NN evaluation requires $\Theta(|S|)$ time in high-dimensional metric spaces [8] (and possibly Euclidean space as well [9]) — a phenomenon known as the algorithmic *curse of dimensionality*. Lastly, the NN classifier has infinite VC-dimension [5], implying that it tends to overfit the data.

This last problem can be mitigated by taking the majority vote among $k > 1$ nearest neighbors [10, 11, 5], or by deleting some sample points so as to attain a larger margin [12].

Shortcomings in the NN classifier led Hart [13] to pose the problem of sample compression. Indeed, significant compression of the sample has the potential to simultaneously address the issues of memory usage, NN search time, and overfitting. Hart considered the minimum Consistent Subset problem — elsewhere called the Nearest Neighbor Condensing problem — which seeks to identify a minimal subset $S^* \subset S$ that is *consistent* with $S$, in the sense that the nearest neighbor in $S^*$ of every $x \in S$ possesses the same label as $x$. This problem is known to be NP-hard [14, 15], and Hart provided a heuristic with runtime $O(n^3)$. The runtime was recently improved by [16] to $O(n^2)$, but neither paper gave performance guarantees.

The Nearest Neighbor Condensing problem has been the subject of extensive research since its introduction [17, 18, 19]. Yet surprisingly, there are no known approximation algorithms for it — all previous results on this problem are heuristics that lack any non-trivial approximation guarantees. Conversely, no strong hardness-of-approximation results for this problem are known, which indicates a gap in the current state of knowledge.

**Main results.** Our contribution aims at closing the existing gap in solutions to the Nearest Neighbor Condensing problem. We present a simple near-optimal approximation algorithm for this problem, where our only structural assumption is that the points lie in some metric space. Define the *scaled margin* $\gamma < 1$ of a sample $S$ as the ratio of the minimum distance between opposite labeled points in $S$ to the diameter of $S$. Our algorithm produces a consistent set $S' \subset S$ of size $\lceil 1/\gamma \rceil^{\mathrm{ddim}(S)+1}$ (Theorem 1), where $\mathrm{ddim}(S)$ is the doubling dimension of the space $S$. This result can significantly speed up evaluation on test points, and also yields sharper and simpler generalization bounds than were previously known (Theorem 3).

To establish optimality, we complement the approximation result with an almost matching hardness-of-approximation lower-bound. Using a reduction from the Label Cover problem, we show that the Nearest Neighbor Condensing problem is NP-hard to approximate within factor $2^{(\mathrm{ddim}(S)\log(1/\gamma))^{1-o(1)}}$ (Theorem 2). Note that the above upper-bound is an absolute size guarantee, and stronger than an approximation guarantee.

Additionally, we present a simple heuristic to be applied in conjunction with the algorithm of Theorem 1, that achieves further sample compression. The empirical performances of both our algorithm and heuristic seem encouraging (see Section 4).

**Related work.** A well-studied problem related to the Nearest Neighbor Condensing problem is that of extracting a small set of simple conjunctions consistent with much of the sample, introduced by [20] and shown by [21] to be equivalent to minimum Set Cover (see [22, 23] for further extensions). This problem is monotone in the sense that adding a conjunction to the solution set can only increase the sample accuracy of the solution. In contrast, in our problem the addition of a point of $S$ to $S^*$ can cause $S^*$ to be inconsistent — and this distinction is critical to the hardness of our problem.

Removal of points from the sample can also yield lower dimensionality, which itself implies faster nearest neighbor evaluation and better generalization bounds. For metric spaces, [24] and [25] gave algorithms for dimensionality reduction via point removal (irrespective of margin size).

The use of doubling dimension as a tool to characterize metric learning has appeared several times in the literature, initially by [26] in the context of nearest neighbor classification, and then in [27] and [28]. A series of papers by Gottlieb, Kontorovich and Krauthgamer investigate doubling spaces for classification [12], regression [29], and dimension reduction [25].

$k$**-nearest neighbor.** A natural question is whether the Nearest Neighbor Condensing problem of [13] has a direct analogue when the 1-nearest neighbor rule is replaced by a $(k > 1)$-nearest neighbor – that is, when the label of a point is determined by the majority vote among its $k$ nearest neighbors. A simple argument shows that the analogy breaks down. Indeed, a minimal requirement for the condensing problem to be meaningful is that the full (uncondensed) set $S$ is feasible, i.e. consistent with itself. Yet even for $k = 3$ there exist self-inconsistent sets. Take for example the set $S$ consisting of two positive points at $(0, 1)$ and $(0, -1)$ and two negative points at $(1, 0)$ and $(-1, 0)$. Then the 3-nearest neighbor rule misclassifies every point in $S$, hence $S$ itself is inconsistent.

**Paper outline.** This paper is organized as follows. In Section 2, we present our algorithm and prove its performance bound, as well as the reduction implying its near optimality (Theorem 2). We then highlight the implications of this algorithm for learning in Section 3. In Section 4 we describe a heuristic which refines our algorithm, and present empirical results.

## 1.1 Preliminaries

**Metric spaces.** A *metric* $d$ on a set $\mathcal{X}$ is a positive symmetric function satisfying the triangle inequality $d(x,y) \le d(x,z) + d(z,y)$; together the two comprise the metric space $(\mathcal{X}, d)$. The diameter of a set $A \subseteq \mathcal{X}$, is defined by $\mathrm{diam}(A) = \sup_{x,y \in A} d(x,y)$. Throughout this paper we will assume that $\mathrm{diam}(S) = 1$; this can always be achieved by scaling.

**Doubling dimension.** For a metric $(\mathcal{X}, d)$, let $\lambda$ be the smallest value such that every ball in $\mathcal{X}$ of radius $r$ (for any $r$) can be covered by $\lambda$ balls of radius $\frac{r}{2}$. The *doubling dimension* of $\mathcal{X}$ is $\mathrm{ddim}(\mathcal{X}) = \log_2 \lambda$. A metric is *doubling* when its doubling dimension is bounded. Note that while a low Euclidean dimension implies a low doubling dimension (Euclidean metrics of dimension $d$ have doubling dimension $O(d)$ [30]), low doubling dimension is strictly more general than low Euclidean dimension. The following packing property can be demonstrated via a repetitive application of the doubling property: For set $S$ with doubling dimension $\mathrm{ddim}(\mathcal{X})$ and $\mathrm{diam}(S) \le \beta$, if the minimum interpoint distance in $S$ is at least $\alpha < \beta$ then

$$|S| \le \lceil \beta/\alpha \rceil^{\mathrm{ddim}(\mathcal{X})+1} \tag{1}$$

(see, for example [8]). The above bound is tight up to constant factors, meaning there exist sets of size $(\beta/\alpha)^{\Omega(\mathrm{ddim}(\mathcal{X}))}$.

**Nearest Neighbor Condensing.** Formally, we define the Nearest Neighbor Condensing (NNC) problem as follows: We are given a set $S = S_- \cup S_+$ of points, and distance metric $d : S \times S \to \mathbb{R}$. We must compute a minimal cardinality subset $S' \subset S$ with the property that for any $p \in S$, the nearest neighbor of $p$ in $S'$ comes from the same subset $\{S_+, S_-\}$ as does $p$. If $p$ has multiple exact nearest neighbors in $S'$, then they must all be of the same subset.

**Label Cover.** The Label Cover problem was first introduced by [31] in a seminal paper on the hardness of computation. Several formulations of this problem have appeared the literature, and we give the description forwarded by [32]: The input is a bipartite graph $G = (U, V, E)$, with two sets of labels: $A$ for $U$ and $B$ for $V$. For each edge $(u, v) \in E$ (where $u \in U$, $v \in V$), we are given a relation $\Pi_{u,v} \subset A \times B$ consisting of admissible label pairs for that edge. A *labeling* $(f, g)$ is a pair of functions $f : U \to 2^A$ and $g : V \to 2^B \backslash \{\emptyset\}$ assigning a set of labels to each vertex. A labeling *covers* an edge $(u, v)$ if for every label $b \in g(v)$ there is some label $a \in f(u)$ such that $(a, b) \in \Pi_{u,v}$. The goal is to find a labeling that covers all edges, and which minimizes the sum of the number of labels assigned to each $u \in U$, that is $\sum_{u \in U} |f(u)|$. It was shown in [32] that it is NP-hard to approximate Label Cover to within a factor $2^{(\log n)^{1-o(1)}}$, where $n$ is the total size of the input.

**Learning.** We work in the *agnostic* learning model [33, 5]. The learner receives $n$ labeled examples $(X_i, Y_i) \in \mathcal{X} \times \{-1, 1\}$ drawn iid according to some unknown probability distribution $\mathbb{P}$. Associated to any *hypothesis* $h : \mathcal{X} \to \{-1, 1\}$ is its *empirical error* $\widehat{\mathrm{err}}(h) = n^{-1} \sum_{i \in [n]} \mathbb{1}_{\{h(X_i) \ne Y_i\}}$ and *generalization error* $\mathrm{err}(h) = \mathbb{P}(h(X) \ne Y)$.

## 2 Near-optimal approximation algorithm

In this section, we describe a simple approximation algorithm for the Nearest Neighbor Condensing problem. In Section 2.1 we provide almost tight hardness-of-approximation bounds. We have the following theorem:

**Theorem 1.** *Given a point set $S$ and its scaled margin $\gamma < 1$, there exists an algorithm that in time*

$$\min\{n^2, 2^{O(\mathrm{ddim}(S))} n \log(1/\gamma)\}$$

*computes a consistent set $S' \subset S$ of size at most $\lceil 1/\gamma \rceil^{\mathrm{ddim}(S)+1}$.*

Recall that an $\varepsilon$-net of point set $S$ is a subset $S_\varepsilon \subset S$ with two properties:

(i) *Packing.* The minimum interpoint distance in $S_\varepsilon$ is at least $\varepsilon$.

(ii) *Covering.* Every point $p \in S$ has a nearest neighbor in $S_\varepsilon$ *strictly* within distance $\varepsilon$.

We make the following observation: Since the margin of the point set is $\gamma$, a $\gamma$-net of $S$ is consistent with $S$. That is, every point $p \in S$ has a neighbor in $S_\gamma$ strictly within distance $\gamma$, and since the margin of $S$ is $\gamma$, this neighbor must be of the same label set as $p$. By the packing property of doubling spaces (Equation 1), the size of $S_\gamma$ is at most $\lceil 1/\gamma \rceil^{\mathrm{ddim}(S)+1}$. The solution returned by our algorithm is $S_\gamma$, and satisfies the guarantees claimed in Theorem 1.

It remains only to compute the net $S_\gamma$. A brute-force greedy algorithm can accomplish this in time $O(n^2)$: For every point $p \in S$, we add $p$ to $S_\gamma$ if the distance from $p$ to all points currently in $S_\gamma$ is $\gamma$ or greater, $d(p, S_\gamma) \geq \gamma$. See Algorithm 1.

---

**Algorithm 1** Brute-force net construction

---

**Require:** $S$
1: $S_\gamma \leftarrow$ arbitrary point of $S$
2: **for all** $p \in S$ **do**
3:     **if** $d(p, S_\gamma) \geq \gamma$ **then**
4:         $S_\gamma = S_\gamma \cup \{p\}$
5:     **end if**
6: **end for**

---

The construction time can be improved by building a *net hierarchy*, similar to the one employed by [8], in total time $2^{O(\mathrm{ddim}(S))} n \log(1/\gamma)$. (See also [34, 35, 36].) A hierarchy consists of all nets $S_{2^i}$ for $i = 0, -1, \ldots, \lfloor \log \gamma \rfloor$, where $S_{2^i} \subset S_{2^{i-1}}$ for all $i > \lfloor \log \gamma \rfloor$. Two points $p, q \in S_{2^i}$ are *neighbors* if $d(p, q) < 4 \cdot 2^i$. Further, each point $q \in S$ is a *child* of a single nearby *parent* point $p \in S_{2^i}$ satisfying $d(p, q) < 2^i$. By the definition of a net, a parent point must exist. If two points $p, q \in S_{2^i}$ are neighbors ($d(p, q) < 4 \cdot 2^i$) then their respective parents $p', q' \in S_{2^{i+1}}$ are necessarily neighbors as well: $d(p', q') \leq d(p', p) + d(p, q) + d(q, q') < 2^{i+1} + 4 \cdot 2^i + 2^{i+1} = 4 \cdot 2^{i+1}$.

The net $S_{2^0} = S_1$ consists of a single arbitrary point. Having constructed $S_{2^i}$, it is an easy matter to construct $S_{2^{i-1}}$: Since we require $S_{2^{i-1}} \supset S_{2^i}$, we will initialize $S_{2^{i-1}} = S_{2^i}$. For each $q \in S$, we need only to determine whether $d(q, S_{2^{i-1}}) \geq 2^{i-1}$, and if so add $q$ to $S_{2^{i-1}}$. Crucially, we need not compare $q$ to all points of $S_{2^{i-1}}$: If there exists a point $p \in S_{2^i}$ with $d(q, p) < 2^i$, then the respective parents $p', q' \in S_{2^i}$ of $p, q$ must be neighbors. Let set $T$ include only the children of $q'$ and of $q'$'s neighbors. To determine the inclusion of every $q \in S$ in $S_{2^{i-1}}$, it suffices to compute whether $d(q, T) \geq 2^{i-1}$, and so $n$ such queries are sufficient to construct $S_{2^{i-1}}$. The points of $T$ have minimum distance $2^{i-1}$ and are all contained in a ball of radius $4 \cdot 2^i + 2^{i-1}$ centered at $T$, so by the packing property (Equation 1) $|T| = 2^{O(\mathrm{ddim}(S))}$. It follows that the above query $d(q, T)$ can be answered in time $2^{O(\mathrm{ddim}(S))}$. For each point in $S$ we execute $O(\log(1/\gamma))$ queries, for a total runtime of $2^{O(\mathrm{ddim}(S))} n \log(1/\gamma)$. The above procedure is illustrated in the Appendix.

## 2.1 Hardness of approximation of NNC

In this section, we prove almost matching hardness results for the NNC problem.

**Theorem 2.** *Given a set $S$ of labeled points with scaled margin $\gamma$, it is NP-hard to approximate the solution to the Nearest Neighbor Condensing problem on $S$ to within a factor $2^{(\mathrm{ddim}(S) \log(1/\gamma))^{1-o(1)}}$.*

To simplify the proof, we introduce an easier version of NNC called *Weighted* Nearest Neighbor Condensing (WNNC). In this problem, the input is augmented with a function assigning weight to each point of $S$, and the goal is to find a subset $S' \subset S$ of minimum *total weight*. We will reduce Label Cover to WNNC and then reduce WNNC to NNC (with some mild assumptions on the admissible range of weights), all while preserving hardness of approximation. The theorem will follow from the hardness of Label Cover [32].

**First reduction.** Given a Label Cover instance of size $m = |U| + |V| + |A| + |B| + |E| + \sum_{e \in E} |\Pi_E|$, fix large value $c$ to be specified later, and an infinitesimally small constant $\eta$. We create an instance of WNNC as follows (see Figure 1).

1. We first create a point $p_+ \in S_+$ of weight 1.

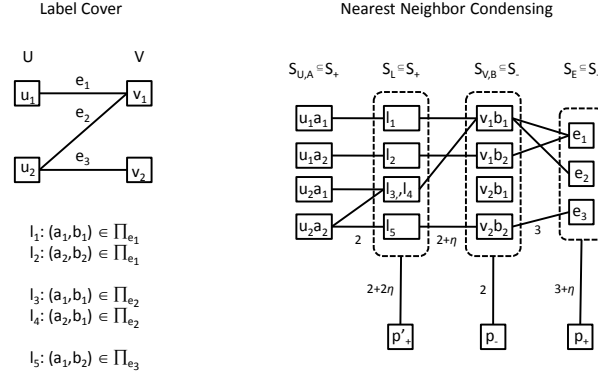

Figure 1: Reduction from Label Cover to Nearest Neighbor Condensing.

We introduce set $S_E \subset S_-$ representing edges in $E$: For each edge $e \in E$, create point $p_e$ of weight $\infty$. The distance from $p_e$ to $p_+$ is $3 + \eta$.

2. We introduce set $S_{V,B} \subset S_-$ representing pairs in $V \times B$: For each vertex $v \in V$ and label $b \in B$, create point $p_{v,b}$ of weight 1. If edge $e$ is incident to $v$ and there exists a label $(a, b) \in \Pi_e$ for any $a \in A$, then the distance from $p_{v,b}$ to $p_e$ is 3.

   Further add a point $p_- \in S_-$ of weight 1, at distance 2 from all points in $S_{V,B}$.

3. We introduce set $S_L \subset S_+$ representing labels in $\Pi_e$. For each edge $e = (u, v)$ and label $b \in B$ for which $(a, b) \in \Pi_e$ (for any $a \in A$), we create point $p_{e,b} \subset S_L$ of weight $\infty$. $p_{e,b}$ represents the set of labels $(a, b) \in \Pi_e$ over all $a \in A$. $p_{e,b}$ is at distance $2 + \eta$ from $p_{v,b}$.

   Further add a point $p'_+ \in S_+$ of weight 1, at distance $2 + 2\eta$ from all points in $S_L$.

4. We introduce set $S_{U,A} \subset S_+$ representing pairs in $U \times A$: For each vertex $u \in U$ and label $a \in A$, create point $p_{u,a}$ of weight $c$. For any edge $e = (u, v)$ and label $b \in B$, if $(a, b) \in \Pi_e$ then the distance from $p_{e,b} \in S_L$ to $p_{u,a}$ is 2.

The points of each set $S_E$, $S_{V,B}$, $S_L$ and $S_{U,A}$ are packed into respective balls of diameter 1. Fixing any target doubling dimension $D = \Omega(1)$ and recalling that the cardinality of each of these sets is less than $m^2$, we conclude that the minimum interpoint distance in each ball is $m^{-O(1/D)}$. All interpoint distances not yet specified are set to their maximum possible value. The diameter of the resulting set is constant, so its scaled margin is $\gamma = m^{-O(1/D)}$. We claim that a solution of WNNC on the constructed instance implies some solution of the Label Cover Instance:

1. $p_+$ must appear in any solution: The nearest neighbors of $p_+$ are the negative points of $S_E$, so if $p_+$ is not included the nearest neighbor of set $S_E$ is necessarily the nearest neighbor of $p_+$, which is not consistent.

2. Points in $S_E$ have infinite weight, so no points of $S_E$ appear in the solution. All points of $S_E$ are at distance exactly $3 + \eta$ from $p_+$, hence each point of $S_E$ must be covered by some point of $S_{V,B}$ to which it is connected – other points in $S_{V,B}$ are farther than $3 + \eta$. (Note that $S_{V,B}$ itself can be covered by including the single point $p_-$.)

   Choosing covering points in $S_{V,B}$ corresponds to assigning labels in $B$ to vertices of $V$ in the Label Cover instance.

3. Points in $S_L$ have infinite weight, so no points of $S_L$ appear in the solution. Hence, either $p'_+$ or some points of $S_{U,A}$ must be used to cover points of $S_L$. Specifically, a point in $S_L \in S_+$ incident on an included point of $S_{V,B} \in S_-$ is at distance exactly $2 + \eta$ from this point, and so it must be covered by some point of $S_{U,A}$ to which it is connected, at distance 2 – other points in $S_{U,A}$ are farther than $2 + \eta$. Points of $S_L$ not incident on an included point of $S_{V,B}$ can be covered by $p'_+$, which at distance $2 + 2\eta$ is still closer than any point in $S_{V,B}$. (Note that $S_{U,A}$ itself can be covered by including a single arbitrary point of $S_{U,A}$, which at distance 1 is closer than all other point sets.)

   Choosing the covering point in $S_{U,A}$ corresponds to assigning labels in $A$ to vertices of $U$ in the Label Cover instance, thereby inducing a valid labeling for some edge and solving the Label Cover problem.

Now, a trivial solution to this instance of WNNC is to take all points of $S_{U,A}, S_{V,B}$ and the single point $p_+$: then $S_E$ and $p_-$ are covered by $S_{V,B}$, and $S_L$ and $p'_+$ by $S_{U,A}$. The size of the resulting set is $c|S_{U,A}| + |S_{U,B}| + 1$, and this provides an upper bound on the optimal solution. By setting $c = m^4 \gg m^3 > m(|S_{U,B}|+1)$, we ensure that the solution cost of WNNC is asymptotically equal to the number of points of $S_{U,A}$ included in its solution. This in turn is exactly the sum of labels of $A$ assigned to each vertex of $U$ in a solution to the Label Cover problem. Label Cover is hard to approximate within a factor $2^{(\log m)^{1-o(1)}}$, implying that WNNC is hard to approximate within a factor of $2^{(\log m)^{1-o(1)}} = 2^{(D \log(1/\gamma))^{1-o(1)}}$.

Before proceeding to the next reduction, we note that to rule out the inclusion of points of $S_E, S_L$ in the solution set, infinite weight is not necessary: It suffices to give each heavy point weight $c^2$, which is itself greater than the weight of the optimal solution by a factor of at least $m^2$. Hence, we may assume all weights are restricted to the range $[1, m^{O(1)}]$, and the hardness result for WNNC still holds.

**Second reduction.** We now reduce WNNC to NNC, assuming that the weights of the $n$ points are in the range $[1, m^{O(1)}]$. Let $\gamma$ be the scaled margin of the WNNC instance. To mimic the weight assignment of WNNC using the unweighted points of NNC, we introduce the following gadget graph $G(w, D)$: Given parameter $w$ and doubling dimension $D$, create a point set $T$ of size $w$ whose interpoint distances are the same as those realized by a set of contiguous points on the $D$-dimensional $\ell_1$-grid of side-length $\lceil w^{1/D} \rceil$. Now replace each point $p \in T$ by twin positive and negative points at mutual distance $\frac{\gamma}{2}$, so that the distance from each twin replacing $p$ to each twin replacing any $q \in T$ is the same as the distance from $p$ to $q$. $G(w, D)$ consists of $T$, as well as a single positive point at distance $\lceil w^{1/D} \rceil$ from all positive points of $T$, and $\lceil w^{1/D} \rceil + \frac{\gamma}{2}$ from all negative points of $T$, and a single negative point at distance $\lceil w^{1/D} \rceil$ from all negative points of $T$, and $\lceil w^{1/D} \rceil + \frac{\gamma}{2}$ from all positive points of $T$.

Clearly, the optimal solution to NNC on the gadget instance is to choose the two points not in $T$. Further, if any single point in $T$ is included in the solution, then all of $T$ must be included in the solution: First the twin of the included point must also be included in the solution. Then, any point at distance 1 from both twins must be included as well, along with its own twin. But then all points within distance 1 of the new twins must be included, etc., until all points of $T$ are found in the solution.

To effectively assign weight to a positive point of NNC, we add a gadget to the point set, and place all negative points of the gadget at distance $\lceil w^{1/D} \rceil$ from this point. If the point is not included in the NNC solution, then the cost of the gadget is only $2$.[2] But if this point is included in the NNC solution, then it is the nearest neighbor of the negative gadget points, and so all the gadget points must be included in the solution, incurring a cost of $w$. A similar argument allows us to assign weight to negative points of NNC. The scaled margin of the NNC instance is of size $\Omega(\gamma/w^{1/D}) = \Omega(\gamma m^{-O(1/D)})$, which completes the proof of Theorem 2.

## 3  Learning

In this section, we apply Theorem 1 to obtain improved generalization bounds for binary classification in doubling spaces. Working in the standard agnostic PAC setting, we take the labeled sample $S$ to be drawn iid from some unknown distribution over $\mathcal{X} \times \{-1, 1\}$, with respect to which all of our probabilities will be defined. In a slight abuse of notation, we will blur the distinction between $S \subset \mathcal{X}$ as a collection of points in a metric space and $S \in (\mathcal{X} \times \{-1, 1\})^n$ as a sequence of point-label pairs. As mentioned in the preliminaries, there is no loss of generality in taking $\text{diam}(S) = 1$. Partitioning the sample $S = S_+ \cup S_-$ into its positively and negatively labeled subsets, the margin induced by the sample is given by $\gamma(S) = d(S_+, S_-)$, where $d(A, B) := \min_{x \in A, x' \in B} d(x, x')$ for $A, B \subset \mathcal{X}$. Any labeled sample $S$ induces the nearest-neighbor classifier $\nu_S : \mathcal{X} \to \{-1, 1\}$ via

$$\nu_S(x) = \begin{cases} +1 & \text{if } d(x, S_+) < d(x, S_-) \\ -1 & \text{else.} \end{cases}$$

We say that $\tilde{S} \subset S$ is $\varepsilon$-*consistent* with $S$ if $\frac{1}{n} \sum_{x \in S} \mathbb{1}_{\{\nu_S(x) \neq \nu_{\tilde{S}}(x)\}} \leq \varepsilon$. For $\varepsilon = 0$, an $\varepsilon$-consistent $\tilde{S}$ is simply said to be *consistent* (which matches our previous notion of consistent subsets). A sample $S$ is said to be $(\varepsilon, \gamma)$-*separable* (with witness $\tilde{S}$) if there is an $\varepsilon$-consistent $\tilde{S} \subset S$ with $\gamma(\tilde{S}) \geq \gamma$.

We begin by invoking a standard Occam-type argument to show that the existence of small $\varepsilon$-consistent sets implies good generalization. The generalizing power of sample compression was independently discovered by [37, 38], and later elaborated upon by [39].

**Theorem 3.** *For any distribution* $\mathbb{P}$, *any* $n \in \mathbb{N}$ *and any* $0 < \delta < 1$, *with probability at least* $1 - \delta$ *over the random sample* $S \in (\mathcal{X} \times \{-1, 1\})^n$, *the following holds:*

*(i) If $\tilde{S} \subset S$ is consistent with S, then* $\quad \mathrm{err}(\nu_{\tilde{S}}) \leq \dfrac{1}{n - |\tilde{S}|} \left( |\tilde{S}| \log n + \log n + \log \dfrac{1}{\delta} \right).$

*(ii) If $\tilde{S} \subset S$ is $\varepsilon$-consistent with S, then* $\quad \mathrm{err}(\nu_{\tilde{S}}) \leq \dfrac{\varepsilon n}{n - |\tilde{S}|} + \sqrt{\dfrac{|\tilde{S}| \log n + 2 \log n + \log \frac{1}{\delta}}{2(n - |\tilde{S}|)}}.$

*Proof.* Finding a consistent (resp., $\varepsilon$-consistent) $\tilde{S} \subset S$ constitutes a *sample compression scheme of size* $|\tilde{S}|$, as stipulated in [39]. Hence, the bounds in (i) and (ii) follow immediately from Theorems 1 and 2 ibid. $\qquad \square$

**Corollary 1.** *With probability at least* $1 - \delta$, *the following holds: If $S$ is $(\varepsilon, \gamma)$-separable with witness $\tilde{S}$, then*

$$\mathrm{err}(\nu_{\tilde{S}}) \quad \leq \quad \frac{\varepsilon n}{n - \ell} + \sqrt{\frac{\ell \log n + 2 \log n + \log \frac{1}{\delta}}{2(n - \ell)}},$$

*where* $\ell = \lceil 1/\gamma \rceil^{\mathrm{ddim}(S)+1}$.

*Proof.* Follows immediately from Theorems 1 and 3(ii). $\qquad \square$

**Remark.** It is instructive to compare the bound above to [12, Corollary 5]. Stated in the language of this paper, the latter upper-bounds the NN generalization error in terms of the sample margin $\gamma$ and $\mathrm{ddim}(\mathcal{X})$ by

$$\varepsilon + \sqrt{\frac{2}{n} \left( d_\gamma \ln(34en/d_\gamma) \log_2(578n) + \ln(4/\delta) \right)}, \tag{2}$$

where $d_\gamma = \lceil 16/\gamma \rceil^{\mathrm{ddim}(\mathcal{X})+1}$ and $\varepsilon$ is the fraction of the points in $S$ that violate the margin condition (i.e., opposite-labeled point pairs less than $\gamma$ apart in $d$). Hence, Corollary 1 is a considerable improvement over (2) in at least three aspects. First, the data-dependent $\mathrm{ddim}(S)$ may be significantly smaller than the dimension of the ambient space, $\mathrm{ddim}(\mathcal{X})$.[3] Secondly, the factor of $16^{\mathrm{ddim}(\mathcal{X})+1}$ is shaved off. Finally, (2) relied on some fairly intricate fat-shattering arguments [40, 41], while Corollary 1 is an almost immediate consequence of much simpler Occam-type results.

One limitation of Theorem 1 is that it requires the sample to be $(0, \gamma)$-separable. The form of the bound in Corollary 1 suggests a natural Structural Risk Minimization (SRM) procedure: minimize the right-hand size over $(\varepsilon, \gamma)$. A solution to this problem was (essentially) given in [12, Theorem 7]:

**Theorem 4.** *Let* $R(\varepsilon, \gamma)$ *denote the right-hand size of the inequality in Corollary 1 and put* $(\varepsilon^*, \gamma^*) = \mathrm{argmin}_{\varepsilon, \gamma} R(\varepsilon, \gamma)$. *Then (i) One may compute* $(\varepsilon^*, \gamma^*)$ *in* $O(n^{4.376})$ *randomized time. (ii) One may compute* $(\tilde{\varepsilon}, \tilde{\gamma})$ *satisfying* $R(\tilde{\varepsilon}, \tilde{\gamma}) \leq 4R(\varepsilon^*, \gamma^*)$ *in* $O(\mathrm{ddim}(S)n^2 \log n)$ *deterministic time. Both solutions yield a witness* $\tilde{S} \subset S$ *of* $(\varepsilon, \gamma)$-separability as a by-product.

Having thus computed the optimal (or near-optimal) $\tilde{\varepsilon}, \tilde{\gamma}$ with the corresponding witness $\tilde{S}$, we may now run the algorithm furnished by Theorem 1 on the sub-sample $\tilde{S}$ and invoke the generalization bound in Corollary 1. The latter holds uniformly over all $\tilde{\varepsilon}, \tilde{\gamma}$.

## 4 Experiments

In this section we discuss experimental results. First, we will describe a simple heuristic built upon our algorithm. The theoretical guarantees in Theorem 1 feature a dependence on the scaled margin $\gamma$, and our heuristic aims to give an improved solution in the problematic case where $\gamma$ is small. Consider the following procedure for obtaining a smaller consistent set. We first extract a net $S_\gamma$ satisfying the guarantees of Theorem 1. We then remove points from $S_\gamma$ using the following rule: for all $i \in \{0, \dots \lceil \log \gamma \rceil\}$, and for each $p \in S_\gamma$, if the distance from $p$ to all opposite labeled points in $S_\gamma$ is at least $2 \cdot 2^i$, then remove from $S_\gamma$ all points strictly within distance $2^i - \gamma$ of $p$ (see Algorithm 2). We can show that the resulting set is consistent:

**Lemma 5.** *The above heuristic produces a consistent solution.*

*Proof.* Consider a point $p \in S_\gamma$, and assume without loss of generality that $p$ is positive. If $d(p, S_\gamma^-) \geq 2 \cdot 2^i$, then the positive net-points strictly within distance $2^i$ of $p$ are closer to $p$ than to any negative point in $S_\gamma$, and are "covered" by $p$. The removed positive net-points at distance $2^i - \gamma$ themselves cover other positive points of $S$ within distance $\gamma$, but $p$ covers these points of $S$ as well. Further, $p$ cannot be removed at a later stage in the algorithm, since $p$'s distance from all remaining points is at least $2^i - \gamma$. ◻

---

**Algorithm 2** Consistent pruning heuristic

---
1: $S_\gamma$ is produced by Algorithm 1 or its fast version (Appendix)
2: **for all** $i \in \{0, \dots, \lceil \log \gamma \rceil\}$ **do**
3:     **for all** $p \in S_\gamma$ **do**
4:         **if** $p \in S_\gamma^\pm$ and $d(p, S_\gamma^\mp) \geq 2 \cdot 2^i$ **then**
5:             **for all** $q \neq p \in S_\gamma$ with $d(p, q) < 2^i - \gamma$ **do**
6:                 $S_\gamma \leftarrow S_\gamma \backslash \{q\}$
7:             **end for**
8:         **end if**
9:     **end for**
10: **end for**

---

As a proof of concept, we tested our sample compression algorithms on several data sets from the UCI Machine Learning Repository. These included the Skin Segmentation, Statlog Shuttle, and Covertype sets.[4] The final dataset features 7 different label types, which we treated as 21 separate binary classification problems; we report results for labels 1 vs. 4, 4 vs. 6, and 4 vs. 7, and these typify the remaining pairs. We stress that the focus of our experiments is to demonstrate that (i) a significant amount of consistent sample compression is often possible and (ii) the compression does not adversely affect the generalization error.

For each data set and experiment, we sampled equal sized learning and test sets, with equal representation of each label type. The $L_1$ metric was used for all data sets. We report (i) the initial sample set size, (ii) the percentage of points retained after the net extraction procedure of Algorithm 1, (iii) the percentage retained after the pruning heuristic of Algorithm 2, and (iv) the change in prediction accuracy on test data, when comparing the heuristic to the uncompressed sample. The results, averaged over 500 trials, are summarized in Figure 2.

| data set | original sample | % after net | % after heuristic | ±% accuracy |
|---|---|---|---|---|
| Skin Segmentation | 10000 | 35.10 | 4.78 | -0.0010 |
| Statlog Shuttle | 2000 | 65.75 | 29.65 | +0.0080 |
| Covertype 1 vs. 4 | 2000 | 35.85 | 17.70 | +0.0200 |
| Covertype 4 vs. 6 | 2000 | 96.50 | 69.00 | -0.0300 |
| Covertype 4 vs. 7 | 2000 | 4.40 | 3.40 | 0.0000 |

Figure 2: Summary of the performance of NN sample compression algorithms.

## Footnotes

[1] A Bayes-consistent modification of the 1-NN classifier was recently proposed in [3].

[2]By scaling up all weights by a factor of $n^2$, we can ensure that the cost of all added gadgets ($2n$) is asymptotically negligible.

[3] In general, $\mathrm{ddim}(S) \leq c \, \mathrm{ddim}(\mathcal{X})$ for some universal constant $c$, as shown in [24].

[4] http://tinyurl.com/skin-data; http://tinyurl.com/shuttle-data; http://tinyurl.com/cover-data

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
