[Reviews · NeurIPS 2014]

Submitted by Assigned_Reviewer_7

The paper studies the problem of finding a small subset S* of the a set S of labelled points
such that the 1-Nearest Neighbour classifier is consistent with S.
The motivation is to speed-up 1NN classification of new points.

The problem of finding a minimal set S* is known to be NP-hard, so the paper
is concerned with approximations. Apparently all the previous results on the problems
concerned heuristics. The present papers presents an algorithm whose approximation
is shown to be optimal, in the sense that doing significantly better is NP-hard.
The approximation quality is in terms of the scaled margin and the doubling dimension.
Some learning-theoretic implications are also outlined.

The algorithm proposed is not very inspiring: it is just constructing an epsilon-net for a specially chosen epsilon.
The proof is also quite simple. However, as far as I see, the main conceptual difficulty was to identify the right
parameters in terms of which to seek an approximation, and this is not trivial.
The technically most difficult result is the optimality, that is, showing that doing significantly
better than the presented algorithm (in terms of the same parameters) is NP-hard.
This is achieved via a reduction to Label Cover.

In sum, I think this is an interesting paper with solid results and should be accepted.

As a minor comment, I find the argument about k nearest neighbours not convincing.
If the labelling is of the set is not self-consistent, the algorithm could be allowed
to just output the empty set. In general, kNN classifiers are much more interesting
than 1NN, since, unlike 1NN, they are universally consistent (in asymptotic) with k=o(n)
for non-deterministic labels.
Summary: This seems to be the first theoretical result on an old problem of finding approximating sets for 1NN classification. The presented algorithm is rather simple but is shown to be nearly optimal. I think one ofthe main contributions is the proper formalization of the problem; as such, the paper is interesting and novel.

Submitted by Assigned_Reviewer_24

This paper presents a new nearest neighbour compression algorithm along with guarantees for its consistency (i.e., the compression does not change the 1-NN labelling of the original points) as well as a bound on the extent of the compression in terms of the margin and doubling dimension for the original set of points. A hardness result is also given showing that compression much beyond what is obtained by the new algorithm is NP-hard. The new algorithm is based on constructing a net for a given sample S of size n which can be done by brute force in O(n^2) or in time linear in n and exponential in the doubling-dimension of S. By leveraging some existing results, the algorithm is shown to enjoy generalisation bounds in the agnostic PAC setting that are improvements over the best, previously known results and are obtained via simpler arguments. Finally, empirical results show that large reductions in sample size can be achieved using the new algorithm and a related heuristic with very little impact on classification accuracy.

This is a very solid piece of work that essentially closes a long outstanding problem in the study of 1-NN classification. The proposed algorithm is elegant and its analysis is clearly presented. To top it off, the algorithm seems very practical and, on the domains tested, very effective.

If there is one criticism I could make, it would be that the paper does not discuss how well the new algorithm compares to the existing techniques in [11] and [14] which do not have theoretical guarantees.
Summary: This is a carefully written, well-motivated, and high quality paper that closes an open problem in the theory of nearest neighbour algorithms by providing a sample compression scheme with consistency and near-optimal performance guarantees. The related work is very well surveyed, results are all clearly stated and appear to be correct, and the empirical results are impressive.

Submitted by Assigned_Reviewer_38

This paper discusses the problem of compressing a set of examples for nearest neighbor (NN) classification. Namely given a set of examples S in some metric space the goal is to output a subset S' of S such that nearest-neighbor labeling of points from S given examples from S' gives the correct labels for all points in S.
It is first observed that if points are separated by at least \gamma then any \gamma-net gives a desired set. Together with standard results for doubling dimension this implies an upper bound of (1/\gamma)^ddim(S) on the size of S'. The authors discuss a greedy algorithm for constructing a \gamma-net and also ways to speed it up and further reduce the number of points heuristically.

In the second result, that is the main technical result they show that the problem is NP-hard to approximate to within a factor of 2^{(\log (1/\gamma)ddim(S))^{1-o(1)}}. NP-hardness of the problem was known but this is a much stronger inapproximability result. It is obtained via an interesting reduction from a label cover.

Finally the authors demonstrate the performance of 1-NN after their sample compression algorithms on several data sets and compare it with using NN directly. The results show that the algorithm does not affect the error of the hypothesis. This is somewhat strange since Occam razor would suggest that there should be perceptible improvement. In either case smaller sample also speeds up NN classification so applying the compression could be useful.

I think that sample compression for NN classification is a natural aspect of this fundamental ML technique and this work makes an interesting contribution to understanding of the problem. Having a nearly optimal lower bound is particularly nice. The techniques are combinations of standard tools and there are no particular surprises here. The paper is also well-written.

Some additional comments:
line 139: agnostic learning is defined for a class of functions. What you describe is just a basic setup for binary classification from iid examples.
line 240: I do not see a definition of c (the weight)
Sec 2: it would be useful to discuss whether the problem remains hard in the Euclidean space
line 362: you suggest that ddim(X) is larger than ddim(S). This does not seem clear to me.
Summary: I think that sample compression for NN classification is a natural aspect of this fundamental ML technique and this work makes an interesting contribution to understanding of the problem.
Author Feedback
Author rebuttal: We thank the referees for detailed and helpful comments.

R_24:
"If there is one criticism I could make, it would be that the paper does not discuss
how well the new algorithm compares to the existing techniques in [11] and [14]
which do not have theoretical guarantees."
At least one large-scale survey of the different NN condensing techniques has been
undertaken:
http://arxiv.org/abs/1007.0085
The results indicate that each technique is best suited for some settings but not
others. Since our nearly matching hardness bound shows that (assuming P!=NP), no
algorithm can significantly and consistently improve ours. In the journal version,
we will perform more comprehensive empirical trials, in addition to the preliminary
results reported here.

R_38:
"line 139: agnostic learning is defined for a class of functions. What you describe is
just a basic setup for binary classification from iid examples."
indeed, we will clarify this

"line 240: I do not see a definition of c (the weight)"
The value c is introduced in line 207 (perhaps we should refer to it as
"weight" to avoid confusion), and specified in line 269.

"Sec 2: it would be useful to discuss whether the problem remains hard in the
Euclidean space"
Indeed, we believe that essentially the same construction can be
shown to hold in Euclidean space, and the details are entirely technical.
Due to space constraints, we planned on deferring this to the full journal
version.

"line 362: you suggest that ddim(X) is larger than ddim(S). This does not seem clear
to me."
Indeed, we were being imprecise. In fact, ddim(S) \le c ddim(X) for some universal
constant c, as shown in
http://www.wisdom.weizmann.ac.il/~robi/papers/GK-NearlyDoubling-SIDMA13.pdf
We will include a careful discussion in the revision.

R_7:
"As a minor comment, I find the argument about k nearest neighbours not
convincing.
If the labelling is of the set is not self-consistent, the algorithm could be allowed
to just output the empty set."
We agree, although the empty set is hardly a useful classifier.
Our main point was that in the noiseless/realizable case, the 1-NN sample condensing
problem always has a nontrivial solution (possibly, the entire sample), which is not
the case for k-NN. We will clarify this in the revision.

"In general, kNN classifiers are much more interesting than 1NN, since, unlike 1NN,
they are universally consistent (in asymptotic) with
k=o(n) for non-deterministic labels."
Indeed, until recently, one could only get a Bayes-consistent NN classifier by
taking k=o(n) neighbors. However, recently, a margin-regularized Bayes-consistent
1-NN classifier was proposed:
http://arxiv.org/abs/1407.0208
We will include a brief discussion in the revision.